# Predicting the Consistency of Vestibular Schwannoma and Its Implication in the Retrosigmoid Approach: A Single-Center Analysis

**DOI:** 10.3390/curroncol32110647

**Published:** 2025-11-19

**Authors:** Raffaele De Marco, Giovanni Morana, Silvia Sgambetterra, Federica Penner, Antonio Melcarne, Diego Garbossa, Michele Lanotte, Roberto Albera, Francesco Zenga

**Affiliations:** 1Department of Neuroscience “Rita Levi Montalcini”, University of Turin, 10124 Turin, Italy; 2Pituitary and Skull Base Surgery Unit, “Città della Salute e della Scienza” University Hospital, 10124 Turin, Italy; 3Neuroradiology Unit, “Città della Salute e della Scienza” University Hospital, 10124 Turin, Italy; 4Neurosurgery Unit, “Città della Salute e della Scienza” University Hospital, 10124 Turin, Italy; 5Division of Otorhinolaryngology, Department of Surgical Sciences, University of Turin, 10126 Turin, Italy

**Keywords:** acoustic neuroma, facial nerve, outcome, magnetic resonance imaging, apparent coefficient diffusion

## Abstract

**Simple Summary:**

Vestibular schwannomas (VSs) are benign brain tumors that can affect hearing and facial nerve function. Their surgical removal can be challenging, as tumors can vary in texture and consistency. These characteristics act as mechanical factors that can influence facial nerve function and impact the patient’s quality of life. Specifically, while firmer tumors may have clearer dissection planes, they could transmit more shock waves to the facial nerve, affecting its electrical response and limiting the extent of resection. Currently, it is challenging to determine a tumor’s consistency prior to surgery. In this study, we examine the role of specific measurements from a standard MRI scan, namely the Apparent Diffusion Coefficient (ADC), in predicting the consistency of VSs. The possibility to easily anticipate tumor consistency allows surgeons to better plan their operations, potentially adjusting their technique to better protect the facial nerve during surgery, which could lead to improved patient outcomes.

**Abstract:**

To explore the relationship between magnetic resonance imaging (MRI) parameters, including T2-weighted intensity and apparent diffusion coefficient (ADC), and intraoperative tumor characteristics, particularly consistency, in vestibular schwannomas (VSs). The association between tumor consistency, facial nerve (FN) function, and postoperative outcomes was analyzed. A single-center retrospective analysis included newly diagnosed VS cases (2020–2023) with cisternal involvement (Samii T3a; volume ≥ 0.7 cm^3^). T2 and ADC maps from the perimetral region of interest were normalized, and tumors were categorized into 3 classes by combining qualitative consistency (soft, fibrous, or fibrous/hard), ultrasonic aspirator power, and adherence to neurovascular structures. FN function was assessed using the House–Brackmann scale at the immediate postoperative period and 12-month follow-up. MRIs of 33 VSs (18 solid and 15 cystic) were analyzed. Normalized values of both T2 (N-T2_mean_) and ADC (N-ADC_min_) maps predicted the classical radiological differentiation. N-ADC_min_ may have some role in predicting consistency (value 1.361, *p* = 0.017, accuracy 0.48) and demonstrated a significant association (*p* = 0.04) with the FN outcome in the immediate postoperative period. An augmented consistency could impair FN function by increasing the intrameatal pressure related to greater transmission of shocks derived from the dissection maneuvers of the cisternal component of the tumor. The possibility of non-invasively exploring VS consistency with a parameter easily calculable on MRI might be beneficial in surgical planning, modifying the timing of the opening of the meatus with respect to what could be the surgical routine in some centers.

## 1. Introduction

Surgical outcomes of vestibular schwannomas (VSs) have often been differentiated into two radiologically defined subtypes: solid (SVS) and cystic (CVS), with the latter usually burdened by worse outcomes. Although it has already been shown that the lack of a standardized definition of CVS [1] led to bias in reports and in the true number of this subtype, a recent meta-analysis [2] has confirmed a worse outcome in terms of anatomical preservation of the facial nerve (FN) and its postoperative function, but not in terms of extent of resection (EOR). More recent studies have questioned this traditional dogma, reporting no significant differences in FN function as well as in EOR rates between CVS and SVS [3,4,5]. A multicenter study found no difference between the two subtypes in terms of EOR and FN outcomes when matched for size and surgical approach [6].

The same meta-analysis [2] yielded a counterintuitive result, revealing an augmented prevalence of adherence to neurovascular structures in the context of SVS, which is a known factor contributing to a worse FN outcome. Similarly, in the early 90s, an increased firmness of the tumor was found to be unfavorable for postoperative FN function, probably due to an increased pressure on the nerve during dissection [7].

The ability to preoperatively predict tumor consistency would therefore be invaluable for surgical planning and patient counseling. Current magnetic resonance imaging (MRI) sequences, such as T2-weighted imaging, offer limited insight into tumor texture. However, advanced diffusion-weighted imaging-derived parameters, specifically the apparent diffusion coefficient (ADC), have emerged as a promising, quantitative tool for characterizing tissue microstructure and density. The ADC value quantifies the random motion of water molecules within tissues; this motion is influenced by cellular density, the integrity of cell membranes, and the viscosity of the extracellular matrix. Consequently, highly cellular or fibrous tissues with restricted extracellular spaces exhibit lower ADC values, while cystic or edematous regions with greater water mobility demonstrate higher ADC values. While other advanced techniques like MR Elastography directly measure biomechanical stiffness, they are not yet routinely available for intracranial neoplasm assessment. In contrast, DWI and ADC maps are now a standard component of most clinical brain MRI protocols, making the ADC value a highly accessible and quantifiable parameter. Initial studies in VS have suggested a potential role for the ADC, with findings indicating higher mean ADC values in cystic compared to solid tumors [8] and worse facial nerve outcomes with low ADC values [9,10]. Considering the contradictory results of the classic solid-cystic dichotomy, the aim of the current work was to investigate a quantifiable and reproducible radiological parameter able to predict the intraoperative consistency of the lesion and to correlate it with surgical results in terms of the extent of resection and rate of facial nerve function preservation.

## 2. Materials and Methods

A single-center retrospective analysis was conducted on patients who underwent resection by a retrosigmoid approach of newly diagnosed vestibular schwannomas in the period 2020–2023. All procedures were conducted by the same skull base surgery team, consisting of neurosurgeons and ear-nose-throat (ENT) surgeons. The surgical technique, the perioperative management, and information regarding the intraoperative neurophysiological monitoring have been described elsewhere [11,12].

The data were collected from medical records and a prospectively maintained database. The Sonopet^®^ ultrasonic aspirator (Stryker Corporation, Kalamazoo, MI, USA) was the only one used during the study period.

The extent of resection was divided as per the latest guidelines [13]. Assessment of the facial nerve function was performed postoperatively (at 3–7 postoperative days before discharge and at the 12-month follow-up visit) based on the House–Brackmann scale [14].

The availability of preoperative MRI, and specifically of T2-weighted sequences (spin, turbo, or fast spin echo-based techniques), diffusion-weighted imaging (DWI) with corresponding Apparent Diffusion Coefficient (ADC) maps, and contrast-enhanced (CE) 3D T1-weighted sequences (both gradient-based and spin echo-based techniques), was considered the main imaging criterion for inclusion in the study. The major diameter of the lesion was calculated on CE 3D T1-weighted sequences, considering only the cisternal component; in addition, its volume (cm^3^) was calculated using the manual segmentation tool offered by BrainLab (BrainLAB AG, Munich, Germany). A differentiation was made between SVS and CVS based on CE T1- and T2-weighted sequences. CVSs were defined according to Piccirillo et al. [15]. The presence of brainstem edema was assessed on T2-weighted sequences and, when available, on fluid-attenuated inversion recovery (FLAIR) images. VSs were also divided considering both the Samii/Hannover and Koos classifications [16,17].

Based on prior studies [18,19], regions of interest (ROIs) were drawn to reproduce as closely as possible the global characteristics of the tumor both on T2-weighted images and on ADC maps; T2 intensity and ADC values (mean, standard deviation (SD), minimum, and maximum values) were calculated and recorded together with other MRI characteristics by an experienced neuroradiologist. After that, these values were divided by an additional ROI drawn on the contralateral middle cerebellar peduncle to obtain normalized mean, SD, min., and max. values (Figure 1). This calculation was conducted on open-source software for visualizing radiological images (https://horosproject.org).

The data regarding the consistency of the lesion were collected by considering the qualitative aspects in the hand of the first surgeon, the range of power of the ultrasonic aspirator (UA) (values expressed as a percentage) used for debulking, and the presence of adhesions reported intraoperatively or as a consequence of a near-total or subtotal resection. Concerning objectivity and reproducibility of the classification, in accordance with others that have been proposed for intracranial meningiomas [18,20], 3 classes were identified as a result of a different association of the three variables (Table 1).

In order to make the data more homogeneous and simultaneously account for any variation regarding the internal auditory canal (IAC) opening (whether before or after debulking), tumor size was considered among the inclusion criteria by selecting only patients with VS occupying the CPA cistern (≥3a according to the Hannover classification). Descriptive statistics were reported with mean and standard deviation for cardinal variables and with frequency and percentage for ordinal and nominal variables. Pearson’s chi-square, Wilcoxon–Mann–Whitney and trend tests were used for categorical variables, while a linear model, ANOVA, Fisher exact test and Student’s *t*-test were used for numerical variables. Analysis of linear regression and binomial or ordinal logistic regression was conducted as well. To evaluate the predictive performance of significant parameters for key binary clinical outcomes, a Receiver Operating Characteristic (ROC) curve analysis was subsequently employed. The area under the curve (AUC) was calculated to quantify discriminative ability, and the *p*-value was obtained from the Wilcoxon Rank Sum Test. A Generalized Linear Model was used to define the optimal cut-off value, considering the value where the sum of specificity and sensitivity was the highest. The DeLong test was applied to compare AUCs. Statistical analysis was performed using an open-source software built on R language (The jamovi project (2021). jamovi. (v2.6) [Computer Software]. Retrieved from https://www.jamovi.org). Statistical significance was set at *p* ≤ 0.05.

## 3. Results

Forty-eight patients underwent surgery for vestibular schwannoma via the retrosigmoid approach during the indicated period of the study. Of these, 15 patients were excluded due to the unavailability of preoperative DWI sequences on MRI studies (*n* = 5), tumor size less than T3a according to Samii’s classification (*n* = 7), and previous radiosurgical treatment by Gamma Knife (*n* = 3). At the end of the selection process, 33 patients (17F:16M) with a mean age of 56.9 ± 13.7 years (range 21.0–80.0) were finally included (Figure A1). Despite the unavailability of T2-weighted sequences, one patient was included as well for ADC calculation. Radiological and intraoperative characteristics were specified in Table A1 and Table A2 (see Appendix A).

The mean value of the normalized T2 intensity (N-T2_mean_) was 2.1 ± 0.4 (range 1.3–3.0) and the mean maximum value (N-T2_max_) was 2.5 ± 0.6 (range 1.5–3.5). In the case of the ADC map calculated for 33 patients, the mean for the normalized mean value (N-ADC_mean_) was 2.0 × 10^−3^ mm/s (SD 0.4, range 1.1–3.3), while the mean for the normalized minimum value (N-ADC_min_) was 1.6 × 10^−3^ mm/s (SD 0.4, range 0.6–2.4). In this case, the minimum value was selected as an expression of the possible compactness and cellularity of the tissue. The difference in all these values in the two classical radiological categories, SVS and CVS, was statistically significant with the exception of N-T2_max_ (Table A3). The thresholds of direct monopolar FN stimulation at the brainstem averaged 0.04 ± 0.03 mA before resection and 0.08 ± 0.08 mA at the end. The delta between the initial and final thresholds was not statistically related to the EOR (*p* = 0.27), although the high number of STRs in the study population tended to be largely ascribed to an excessive impairment of the intraoperative neurophysiological parameters, which, in the presence of difficulty in dissociating the capsule from the FN, consciously led to the interruption of the procedure. Indeed, the presence of continuous EMG discharges, significant variation in corticobulbar potentials, or variation in direct stimulation parameters leads us to interrupt any maneuvers, irrigate with warm saline, and/or use papaverine. After a variable waiting period, if there is no improvement in neurophysiological parameters, we avoid further resection to preserve FN function. There were no intra- or postoperative complications related to the procedure, such as hemorrhage, hydrocephalus, incisional and non-incisional CSF leak, surgical wound infection, meningitis, or death. Considering the facial nerve deficit graded according to the House–Brackmann scale and summarized into only two classes (HB < 3, favorable, and HB ≥ 3, unfavorable), 22 patients belonged to the first class (62.9%) in the immediate postoperative period. These numbers became 28 (80%) at the clinical evaluation 12 months after the procedure. Hearing preservation surgery was not performed because all recruited cases had a more or less severe deficit of hearing function preoperatively (class C and D AAO-HNS (American Academy of Otolaryngology-Head and Neck Surgery).

### 3.1. Predicting the Consistency

Regarding the intraoperative aspects (Table A2), there were 8 VSs (24.2%) of soft consistency with fibrous areas, 16 VSs with predominantly fibrous or intermediate characteristics (48.5%), and 15 (45.5%) with increased consistency, but never calcific as can be found in some meningiomas (therefore classes 1 and 5 of the classification proposed by Zada et al. [20] were not taken into consideration). The ultrasonic aspirator was rarely used at maximum powers (90–100%) to avoid even thermal damage to the FN (Table A2). The Chi-squared test showed a relationship between these two variables (χ^2^ = 21.3, *p* < 0.001) (Figure 2).

Adhesions were associated neither with the subjective consistency of the lesion (*p* = 0.97) nor with UA power (*p* = 0.74). A higher frequency of this aspect was observed in CVS, with an odds ratio of 16.9 (95% CI 2.76–103). Conversely, brainstem edema exhibited no correlation with the presence of adhesions (*p* = 0.12).

In order to obtain a more reliable parameter for measuring the consistency of the lesion, the proposed score was applied (Table 1). As for meningiomas, the degree of tumor capsule mobilization was considered as well as the presence/absence of adhesions with the surrounding structures. This resulted in 8 patients belonging to class A (22.9%), 18 to class B (51.4%), and 9 to class C (25.7%).

As demonstrated in Table 2, a substantial correlation was identified among these consistency classes and tumor size, with a propensity to observe greater consistency in larger tumors. Additionally, a negative association was observed between N-ADC_min_ and the highest class (*p* = 0.03), with lower values corresponding to this category. A similar trend was noted for surgical time (*p* = 0.04), with tumors classified as Class C, which were characterized by a higher prevalence of adhesions, requiring more challenging procedures. Furthermore, a significant correlation (although weak) was found between the extent of resection and the consistency classes (*p* = 0.05).

Linear regression models were applied in order to better characterize these relationships and to investigate the role played by the values collected in the T2-weighted and DWI-ADC sequences (Table 3 and Table 4). Specifically, the models that showed the best values of R, R^2^, and adjusted R^2^ were those in which the tumor volume, the VS type (SVS vs. CVS), the consistency class (B vs. A and C vs. A), the extent of resection (NTR vs. STR and GTR vs. NTR), and the delta of the stimulation thresholds were considered among the variables. These variables were related both to the N-T2_mean_ (Table 2) and to the N-ADC_min_ (Table 3). Passing the normality test (Shapiro–Wilk) for both N-T2_mean_ 0.963 (*p* = 0.304) and N-ADC_min_ 0.975 (*p* = 0.628), the regression models had an R of 0.822, R^2^ of 0.676, and adjusted R^2^ of 0.572 for N-T2_mean_ and 0.795, 0.632, and 0.509 for N-ADC_min_. The Variance Inflation Factor (VIF) for each independent variable was below 1.6, suggesting an acceptable level of collinearity.

Focusing on the linear regression concerning N-ADC_min_ (Table 4), the model was found to be most significantly influenced by the consistency class and the EOR. Furthermore, the contribution of the difference between the stimulation thresholds was more significant compared to the previous analysis. The relationship was negative or inverse in the case of consistency classes, especially with Class C in relation to Class A, where a lower N-ADC_min_ value was more likely to be found (*p* < 0.001).

### 3.2. Association with the Outcome

Once the association with the consistency class for both N-T2_mean_ and N-ADC_min_ had been understood, the possible involvement of these parameters on facial nerve function in the immediate postoperative period (defined as FU0) (Table A4 and Table A5) and one year after the procedure (FU12) (Table A6 and Table A7) was evaluated. The dependent variable was dichotomized into two categories: favorable and unfavorable (<HB3 and ≥HB3). The multivariate analysis revealed a significant positive association between the delta threshold and the dependent variable. A modest association was found for tumor volume, which was on average twice as large among those with an unfavorable FN function (4.7 ± 3.5 vs. 10.6 ± 12.8, *p* = 0.048). Conversely, age, consistency class, and cystic appearance did not reach statistical significance. However, in certain models, the inverse association was observed to be with the extent of resection rather than the delta.

Under the hypothesis that a higher consistency class might be associated with a higher FN manipulation (also due to the presence of adhesions), a more detailed analysis was conducted on the relationship between the delta of the stimulation thresholds (T0 and T1) and the consistency classes. Although the association was not statistically significant (Kruskal–Wallis test = 0.935, *p* = 0.63), an upward trend of the mean of these differences was seen: 0.035 mA in class A, 0.047 in class B, and 0.052 in class C (Figure 3).

In the ROC analysis for predicting an unfavorable immediate FN outcome (HB ≥ 3), N-ADC_min_ showed a modest predictive capacity, with an AUC of 0.72 (95% CI: 0.52–0.93, *p* = 0.006), slightly superior if compared to the N-T2_mean_ value. The optimal cut-off value was 1.417 × 10^−3^ mm^2^/s, yielding a sensitivity of 72.7% and a specificity of 81.0% (Figure 4).

## 4. Discussion

The classic solid versus cystic dichotomy pervades the literature on vestibular schwannomas. CVSs are typically described as characterized by a larger mean diameter, rapid growth, and acute clinical presentation [21,22]. The rapid growth with a sudden displacement of leptomeninges has been advanced as one possible cause of an increased percentage of adherence between the tumor and the surrounding neurovascular structures [23]. This makes resection and functional preservation of the facial nerve more challenging, with only a few studies reporting a higher rate of unfavorable outcome (HB ≥ 3) in SVS compared to CVS [24,25]. Recent results from large sample sizes have shown no significant difference in the extent of resection and facial nerve function [5,6,26]. Instead, the factors that influence these outcomes seem to be the surgical approach and the size of the tumor. Indeed, the latter is an unfavorable factor for cystic subtypes, as they are generally larger at diagnosis [26].

Although our analysis did not reveal a statistically significant association between the VS type and the postoperative facial nerve function, it was more likely to observe an impaired facial nerve function preoperatively in CVSs (*p* = 0.04), supporting the hypothesis of sudden pathogenetic mechanisms underlying cyst formation, and a higher rate of adherence (*p* < 0.01), resulting in an increased risk of nerve injury during dissection maneuvers. Beyond the radiological definition standardized in 2003 [27] and then refined by Piccirillo’s group [15], certain radiological features may anticipate intraoperative aspects. Indeed, imaging modalities such as T2-weighted sequences [28], diffusion weighted imaging [29], and spectroscopy [30] on brain MRI, or more recently MR elastography, MRE [31] have been instrumental in differentiating between soft and fibrous tumors, specifically in meningiomas. MRE uniquely quantifies tissue stiffness by measuring the propagation of mechanically induced shear waves, providing a direct, quantitative measure of a tumor’s viscoelastic properties [32,33]. Duhon et al. have recently validated MRE’s superior accuracy in predicting the intraoperative consistency of VS and meningiomas, showing strong correlations with surgeon assessment and ultrasonic aspirator settings [33]. Furthermore, MRE-derived stiffness has been directly linked to critical surgical outcomes: stiffer VSs are associated with worse preoperative hearing, a higher likelihood of subtotal resection, and poorer immediate and long-term facial nerve function [33]. However, this technique is not without limitations: for example, it loses reliability in the case of small or intracanalicular VS; it also requires specialized hardware, bespoke sequences, and complex post-processing, making it less universally available than the ADC derived from routine clinical DWI. The ADC quantifies water molecule movement within the tumor tissue, reflecting its microstructural properties. Lower ADC values indicate denser, more fibrous tumors with less extracellular water, whereas higher ADC values correspond to softer, more cystic tumors [34]. In general, a lower ADC value corresponds to a higher cellularity value or a rather low water content. Although with conflicting results in the literature, sometimes opposite when other tumors are taken into account [35], in the case of intracranial tumors, its diagnostic value has been demonstrated in predicting the grade of gliomas [36] and meningiomas [37,38] (lower value in high-grade gliomas and atypical/anaplastic meningiomas). In vestibular schwannomas, Chuang et al. noticed a higher mean ADC value in CVS [39] and a similar result came out in the current analysis (1.7 ± 0.3 for SVS vs. 2.3 ± 0.4 for CVS, *p* < 0.01), although the difference was less significant. Despite the absence of objectivity and reproducibility in the definitions of hypo- or hyperintensity on T2-weighted sequences and intraoperative consistency [18], a select group of researchers have directed their efforts toward the analysis of T2-weighted sequences. As posited by Copeland et al. [40], VSs characterized as “soft” exhibited a higher propensity to be hyperintense on T2-weighted sequences (88% vs. 14%, *p* < 0.005), while those designated as “firm” were more likely to be hypointense on T2-weighted sequences (86% vs. 6%, *p* < 0.005). A greater tendency towards an unfavorable outcome (HB ≥ 3) was identified for the “firm” ones, but this was not statistically significant (43% versus 19%, *p* = 0.14). A similar result was confirmed in another analysis by Rizk et al. [25], although the difference was significant only in the immediate postoperative period. Our findings demonstrated a correlation between N-T2_mean_ and the presence of cysts, irrespective of their type (A or B according to Piccirillo et al. [15]) (*p* < 0.001), as well as with EOR. The latter association may be more clearly elucidated through the examination of the inverse relationship between N-T2_mean_ and consistency classes. Specifically, lower N-T2_mean_ levels were more frequently observed in higher classes (*p* < 0.01). The mean value of N-T2_mean_ was lower in HB ≥ 3 patients than in those with a favorable FN outcome, both perioperatively (2.2 vs. 1.9, *p* = 0.11) and at follow-up (2.2 vs. 1.8, *p* = 0.08). Nevertheless, this aspect did not attain statistical significance. The intraoperative presence of adhesions was involved in the immediate FN function (*p* < 0.03), but not in the long-term (*p* = 0.22). In regard to operative time, Macielak et al. [19] identified a positive correlation with the T2-w intensity ratio, with a higher ratio indicative of a more time-consuming operation due to a more complex dissection in the absence of a favorable plane. Another study group suggested a correlation between the ADC value and adverse FN outcomes, based on identified cut-offs for this continuous variable, no matter what approach was used [9,10]. In their results, the intraoperative characteristics of consistency (obtained only from the descriptive analysis from the operative reports) and adhesions were associated with poor HB, particularly if firmer and with more adhesions. Despite the modest patient sample size (see limitations), the approach of reporting consistency in a semi-objective manner exhibited a substantial association with this radiological parameter (*p* = 0.027). However, it did not demonstrate a significant association with postoperative facial nerve function (*p* = 0.11). Nonetheless, a higher mean N-ADC_min_ (1.70 × 10^−3^ vs. 1.45 × 10^−3^) was documented among the favorable HB results. Indeed, in contrast to the possibility of predicting the facial nerve outcome (which is a multifaceted process involving numerous factors, such as tumor size, location, surgical technique, and intraoperative management [11,12]), this radiological parameter may offer a more explanatory perspective in predicting what the surgeon will face intraoperatively, such as consistency and/or adherence with the neurovascular structures.

## 5. Limitations

The study’s retrospective nature and inclusion of multiple VS size classes limit results. The final sample size (*n* = 33), while informative, is modest and reduces the statistical power of the analyses by increasing the risk of overfitting in our regression models. A significant statistical limitation arises from the imbalanced distribution of tumors across consistency classes, with a concentration of cases in the fibrous category (Class B). Such an imbalance can skew a predictive model and make accuracy an ineffective performance measure. Furthermore, the classification of tumor consistency, while improved by combining qualitative assessment, UA power, and adherence, retains inherent subjective elements. The initial qualitative descriptor (e.g., “soft with fibrous areas”) is based on the surgeon’s tactile feedback, which is difficult to standardize. The use of UA power as an objective metric is also susceptible to bias; for instance, surgeons may deliberately use lower power on smaller, firmer tumors due to their close proximity to critical neurovascular structures, which could misclassify their consistency. Conversely, higher power might be used on larger, softer tumors simply to expedite debulking. Although all procedures were performed by the same skull base team and mostly by a single surgeon, which standardizes the approach, we did not control for potential confounders such as individual surgeon experience or subtle variations in technique. To confirm these results, information from quantifiable pathological elements (e.g., the degree of collagen at hematoxylin/eosin fixation [41]) or tools like elastography probes (e.g., the ultrasound with an elastography module, although there are still no probes capable of this method small enough to be used in the RS approach [42]) is necessary. The main strength of the study lies in the identification of a parameter that offers the possibility of rapidly acquiring objective data that can be easily and quickly obtained, even directly by the surgeon, from sequences that are constantly present in MRI scans. Increasing the number of patients, tumor volumes, and VS types, and validating the findings, will strengthen the study’s results.

## 6. Conclusions

The possibility of better defining a VS on preoperative brain MRI through an easily calculable parameter such as ADC could lead to an evolution of the classic “solid vs. cystic” dichotomy. The association between increased consistency or lower N-ADC_min_ and worse immediate FN function may support the hypothesis of increased shock transmission to the facial nerve during dissection maneuvers. In addition to transmission or greater manipulation, one could also hypothesize a greater increase in intrameatal pressure. Based on this, a hypothetical surgical strategy can be proposed. In centers where the internal auditory meatus is typically opened after initial tumor debulking, a preoperative prediction of firm consistency might justify an early opening of the meatus in eligible tumors (e.g., Koos 2 and select Koos 3), potentially mitigating pressure on the nerve. However, this recommendation remains speculative and must be validated by future, larger studies specifically designed to test this intraoperative strategy.

## Figures and Tables

**Figure 1 curroncol-32-00647-f001:**
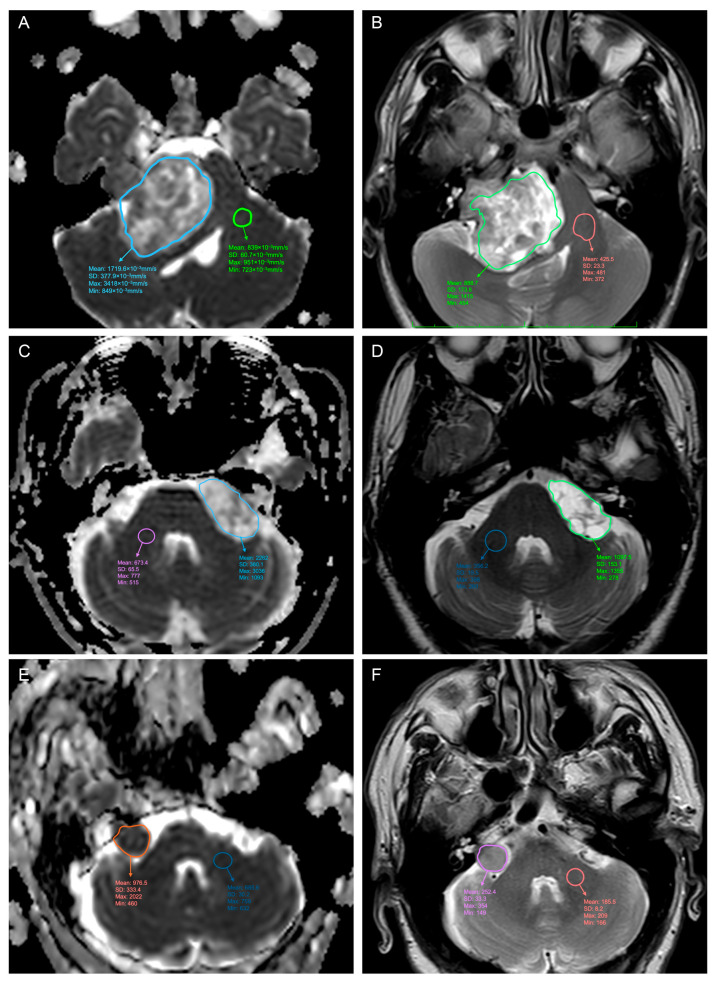
Perimetral regions of interest drawings of three different Vestibular Schwannomas in their cisternal part, respectively, on ADC (left; **A**,**C**,**E**) and T2-weighted sequences (right; **B**,**D**,**F**). Another ROI was drawn on the contralateral middle cerebellar peduncle for normalization. Rather than a circular ROI encompassing only a small part of the tumor or only a solid part of it, an ROI was drawn using the ‘pencil’ tool on Horos along the perimeter of the tumor, selecting only the cisternal part at its point of maximum extension.

**Figure 2 curroncol-32-00647-f002:**
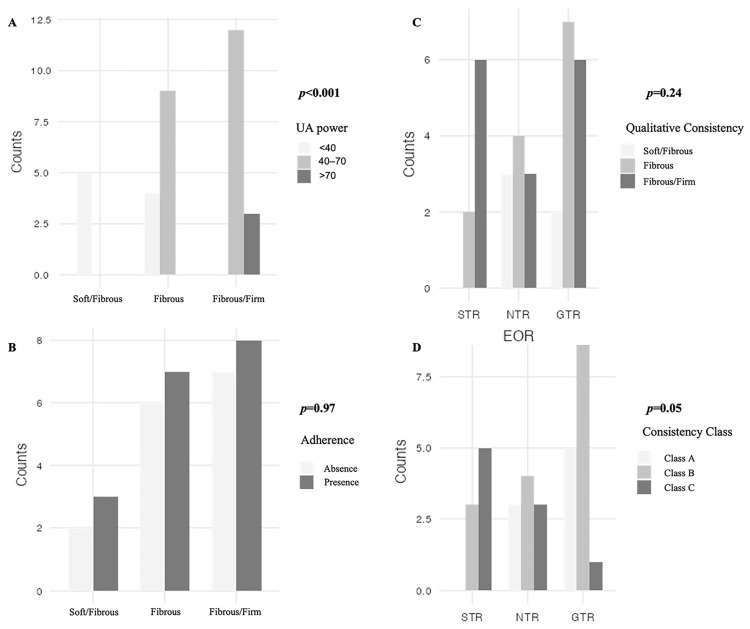
Distribution of different intraoperative characteristics. A simple Chi-squared independent test was applied. In (**A**), a significant difference was found among the classes of intraoperative subjective consistency (soft/fibrous, fibrous and fibrous/firm) and the use of different power range (χ^2^ = 21.3, *p* < 0.001), with higher power mostly seen in firmer VS; the distribution of adherence which were recognized intraoperatively, did not differ significantly between qualitative consistency classes χ^2^ = 0.07, *p* = 0.97 (as shown in (**B**)); the subjective experience on tumor firmness (qualitative consistency) did not showed a relationship with the extent of resection χ^2^ = 5.48, *p* = 0.24 (**C**), but the class of consistency—qualitative consistency + UA power range + presence/absence of adherences—showed a modest difference in distribution between EOR, χ^2^ = 9.35, *p* = 0.05 (**D**).

**Figure 3 curroncol-32-00647-f003:**
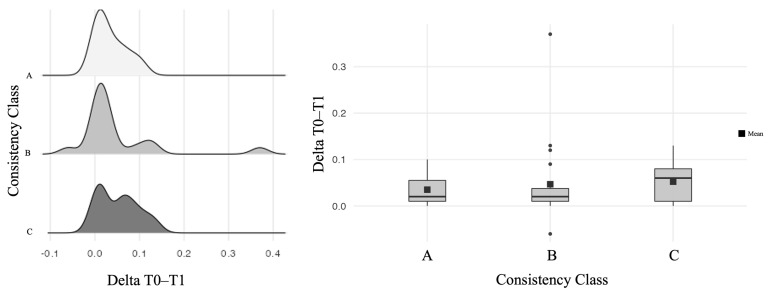
Density histograms of the distribution of delta thresholds (T0–T1) among classes of consistency (**left**) and box plots (**right**) that show a different distribution of the mean (although not statistically significant) of this parameter among classes of consistency: 0.035 in class A, 0.024 in class B, and 0.052 in class C.

**Figure 4 curroncol-32-00647-f004:**
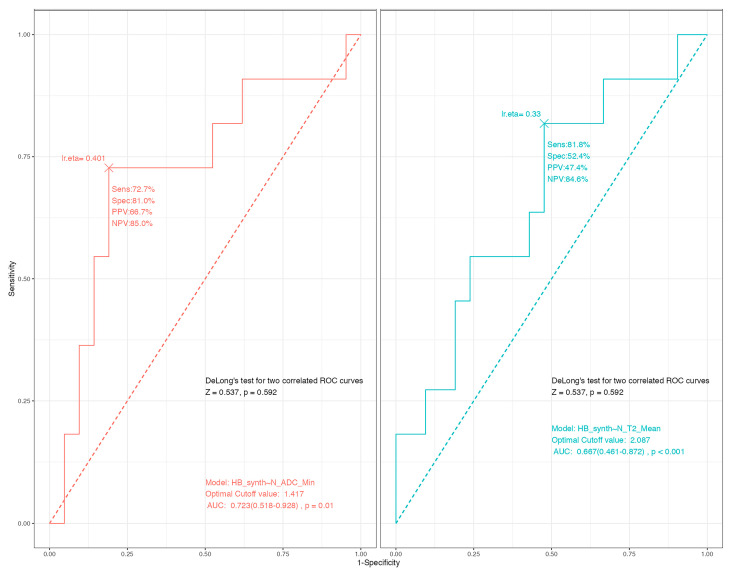
Receiver Operating Characteristic (ROC) curves for predicting unfavorable immediate facial nerve outcome (HB grade ≥ 3). This faceted plot compares the predictive performance of the normalized minimum Apparent Diffusion Coefficient (N-ADC_min_) and the normalized mean T2-weighted intensity (N-T2_mean_). The red curve (**left**) represents N-ADC_min_, which demonstrated an Area Under the Curve (AUC) of 0.72 (95% CI: 0.52–0.93, *p* = 0.006), indicating a modest predictive capacity. The blue curve (**right**) represents N-T2_mean_. The optimal cut-off value for N-ADC_min_ was determined to be 1.417 × 10^−3^ mm^2^/s, yielding a sensitivity of 72.7% and a specificity of 81.0%. The abbreviation “lr.eta” refers to the logistic regression estimated value associated with this optimal cut-off point. A DeLong’s test for comparing the two AUCs showed no statistically significant difference (*p* = 0.59), suggesting that while N-ADC_min_ had a numerically higher AUC, its performance was not significantly superior to N-T2_mean_ in this cohort.

**Table 1 curroncol-32-00647-t001:** Consistency Class Score System.

Intraoperative Consistency (Qualitative)	Ultrasonic Aspirator (UA) Power Range	Adherence
Soft with fibrous areas (1)	<40 (1)	Yes (1)
Fibrous (2)	40–70 (2)	No (0)
Fibrous/Firm (3)	>70 (3)	
**Class A ≤ 3**	**Class B = 4–5**	**Class C = 6–7**

**Table 2 curroncol-32-00647-t002:** Relationships (Univariate analysis) between radiological and intraoperative characteristics and the consistency classes. ^1^ Pearson’s Chi-squared test and ^2^ Linear Model ANOVA.

Radiological and Intraoperative Characteristics	Consistency Class				
	Class A (N = 8)	Class B (N = 16)	Class C (N = 9)	Total (N = 33)	*p* Value
**Max Diameter**					**0.027 ^2^**
Mean (SD)	20.8 (6.3)	21.7 (8.0)	30.2 (9.1)	23.8 (8.7)	
Range	14.0–33.0	12.0–40.0	22.0–48.6	12.0–48.6	
**Tumor Volume**					**0.015 ^2^**
Mean (SD)	3.9 (3.0)	4.4 (4.7)	13.2 (12.4)	6.7 (8.2)	
Range	0.8–8.7	0.7–18.5	3.8–38.3	0.7–38.3	
**Koos Grade**					**0.017 ^1^**
2	3.0 (37.5%)	7.0 (43.8%)	0.0 (0.0%)	10.0 (30.3%)	
3	5.0 (62.5%)	4.0 (25.0%)	3.0 (33.3%)	12.0 (36.4%)	
4	0.0 (0.0%)	5.0 (31.2%)	6.0 (66.7%)	11.0 (33.3%)	
**Samii Classification**					**0.002 ^1^**
T3a	3.0 (37.5%)	7.0 (43.8%)	0.0 (0.0%)	10.0 (30.3%)	
T3b	5.0 (62.5%)	4.0 (25.0%)	1.0 (11.1%)	10.0 (30.3%)	
T4a	0.0 (0.0%)	4.0 (25.0%)	5.0 (55.6%)	9.0 (27.3%)	
T4b	0.0 (0.0%)	1.0 (6.2%)	3.0 (33.3%)	4.0 (12.1%)	
**N-T2_mean_**					0.068 ^2^
N-Miss	1.0	0.0	0.0	1.0	
Mean (SD)	2.4 (0.4)	2.1 (0.5)	1.9 (0.3)	2.1 (0.4)	
Range	1.9–3.0	1.4–2.9	1.3–2.3	1.3–3.0	
**N-ADC_min_**					**0.027 ^2^**
Mean (SD)	1.9 (0.4)	1.7 (0.4)	1.3 (0.4)	1.6 (0.4)	
Range	1.4–2.4	1.0–2.3	0.6–1.9	0.6–2.4	
**Operative Time (min)**					0.066 ^2^
Mean (SD)	356.2 (66.6)	374.9 (111.2)	462.2 (100.3)	394.2 (105.5)	
Range	250–445	210–630	300–630	210–630	
**EOR**					**0.015 ^1^**
STR	0.0 (0.0%)	3.0 (18.8%)	5.0 (55.6%)	8.0 (24.2%)	
NTR	3.0 (37.5%)	4.0 (25.0%)	3.0 (33.3%)	10.0 (30.3%)	
GTR	5.0 (62.5%)	9.0 (56.2%)	1.0 (11.1%)	15.0 (45.5%)	

**Table 3 curroncol-32-00647-t003:** Multivariate linear regression for N-T2w_mean_.

Linear Regression Dependent Variable N-T2w_mean_							95% CI	
Predictor		Estimate	SE	t	*p*	Stand. Estimate	Lower	Upper
**Intercept ^a^**		1.90600	0.35108	5.429	<0.001			
**VS Type**	**CVS–SVS**	0.51126	0.13049	3.918	**<0.001**	1.1395	0.5405	1.739
**Tumor Volume**		0.00792	0.00666	1.189	0.246	0.1771	−0.1296	0.484
**Consistency Class**	**B–A**	−0.42070	0.13845	−3.039	**0.006**	−0.9377	−1.5733	−0.302
	**C–A**	−0.55195	0.17175	−3.214	**0.004**	−1.2302	−2.0186	−0.442
**Operative time**		1.47 × 10^−4^	6.59 × 10^−4^	0.223	0.825	0.0356	−0.2927	0.364
**EOR**	**NTR–STR**	0.33901	0.16108	2.105	**0.046**	0.7556	0.0162	1.495
	**GTR–STR**	0.29739	0.17700	1.680	0.105	0.6629	−0.1496	1.475
**Delta T0–T1**		−0.82178	0.82024	−1.002	0.326	−0.1314	−0.4014	0.139

^a^ Represents reference level.

**Table 4 curroncol-32-00647-t004:** Multivariate linear regression for N-ADC_min_.

Linear regression for N-ADC_min_							95% CI	
Predictor		Estimate	SE	t	*p*	Stand. Estimate	Lower	Upper
**Intercept ^a^**		0.64636	0.3601	1.795	0.085			
**VS Type**	**CVS–SVS**	0.63426	0.1565	4.052	**<0.001**	1.484	0.7284	2.2406
**Tumor Volume**		−0.00639	0.0101	−0.631	0.534	−0.123	−0.5236	0.2783
**Consistency Class**	**B–A**	−0.26317	0.1373	−1.916	0.067	−0.616	−1.2794	0.0475
	**C–A**	−0.66921	0.1733	−3.861	**<0.001**	−1.566	−2.4037	−0.7289
**Operative time**		0.00159	7.36 × 10^−4^	2.162	**0.041**	0.393	0.0178	0.7679
**EOR**	**NTR–STR**	0.22648	0.1707	1.327	0.197	0.530	−0.2945	1.3547
	**GTR–STR**	0.43400	0.1936	2.242	**0.034**	1.016	0.0806	1.9510
**Delta T0–T1**		4.19394	1.6463	2.547	**0.018**	0.426	0.0810	0.7720

^a^ Represents reference level.

## Data Availability

The raw data supporting the conclusions of this article will be made available by the authors on request.

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
