# Peer review of "Predicting the Consistency of Vestibular Schwannoma and Its Implication in the Retrosigmoid Approach: A Single-Center Analysis"

_curroncol, 2025, doi:10.3390/curroncol32110647_

Round 1
Reviewer 1 Report
Comments and Suggestions for Authors
The study addresses a practical gap in VS surgery: Preoperative prediction of tumor consistency, which influences dissection difficulty, extent of resection, and FN preservation. The manuscript is well-structured, with clear objectives, methods, and discussions, but it requires minor revisions for clarity, expanded limitations, and tempered conclusions.
Limitations and weaknesses: Retrospective design introduces selection bias (e.g., only 33/48 cases included due to missing DWI; excluded small tumors. Small sample (n=33; 18 solid, 15 cystic), reducing statistical power and risking overfitting in regressions. Subjective elements in consistency scoring (e.g., "soft with fibrous areas" relies on the surgeon's hand feel; UA power may vary by tumor size/proximity to structures). No control for confounders (e.g., surgeon experience, even though it's the same team).
I suggest discussing potential bias from UA power variability (e.g., lower power in small tumors despite firmness). Use ROC curves to define N-ADCmin cut-offs for consistency/FN prediction. Expand limitations and soften surgical recommendations as "hypothetical."
Author Response
Comment 1: The study addresses a practical gap in VS surgery: Preoperative prediction of tumor consistency, which influences dissection difficulty, extent of resection, and FN preservation. The manuscript is well-structured, with clear objectives, methods, and discussions, but it requires minor revisions for clarity, expanded limitations, and tempered conclusions.
Response 1: Thank you for taking the time to review our manuscript and for your positive comments. We appreciate the opportunity to improve the quality of our work.
Comment 2: Limitations and weaknesses: Retrospective design introduces selection bias (e.g., only 33/48 cases included due to missing DWI; excluded small tumors. Small sample (n=33; 18 solid, 15 cystic), reducing statistical power and risking overfitting in regressions. Subjective elements in consistency scoring (e.g., "soft with fibrous areas" relies on the surgeon's hand feel; UA power may vary by tumor size/proximity to structures). No control for confounders (e.g., surgeon experience, even though it's the same team).
I suggest discussing potential bias from UA power variability (e.g., lower power in small tumors despite firmness).
Response 2: Thank you for your comments and suggestions. We tried to emphasize even more the limitations, which were discussed in an unusually long section, aware of the fact that a retrospective study with few patients cannot be an adequate sample to be able to say anything definitive.
Comment 3: Use ROC curves to define N-ADCmin cut-offs for consistency/FN prediction.
Response 3: It was not possible to conduct the ROC analysis on the consistency classes, as there were only three of them. Nevertheless, We performed this analysis considering N-ADCmin alone and together with N-T2mean and their ability to predict the FN outcome. We reported these new data in the results (lines 264-267, page 9) and a new figure (Figure 4).
Comment 4: Expand limitations and soften surgical recommendations as "hypothetical."
Response 4: Thank you for your comment. We have softened our conclusions in light of the limitations that affect the validity of our results.
Reviewer 2 Report
Comments and Suggestions for Authors
nice work about the consistency of vestibular schwannoma and its
implication in the retrosigmoid approach
pleas relook with care the english language used
add more elements in the introduction of the study
figure 1 is very nice please add more rediological images and some clinical images
in order to support better the text
table 1 is ok
table a1 about demographics is nice but huge
please split the informations
table a2 is ok
in figure 2 please minimize the maths used in order to be more easy for the medical audience to read it and to understeand it
same comments for tables 2 and a3 please ninimize the maths used-if possible spilt them or add some graphics
add more elements in discussion
add more and more recent references
add more and more detailed conclusions
mention the innovantions of this study in the introduction
add in the conclusion
Comments on the Quality of English Languagethe english language can be improved
Author Response
Comment 1: nice work about the consistency of vestibular schwannoma and its
implication in the retrosigmoid approach
Response 1: Thank you for taking the time to review our manuscript and for your comments. We appreciate the opportunity to improve the quality of our work. We have tried to answer point-by-point.
Comment 2: pleas relook with care the english language used
Response 2: An English mother tongue has reviewed the manuscript. Different sentences have been reformulated for a better understanding. These are reported in red in the revised version of the manuscript.
Comment 3: add more elements in the introduction of the study
Response 3: Thank for your comment. We added more data on the background.
Comment 4: figure 1 is very nice please add more rediological images and some clinical images
Response 4: We added two more examples. Regarding clinical images, if you mean pictures of patients’ face we usually do not store any.
in order to support better the text
table 1 is ok
Comment 5: table a1 about demographics is nice but huge
please split the informations
Response 5: Thank for your comment. We are aware of the amount of information of this table. In fact, we intentionally report it in the appendix as supplementary material.
table a2 is ok
Comment 6: in figure 2 please minimize the maths used in order to be more easy for the medical audience to read it and to understeand it
Response 6: Thank you for pointing this out. The image does not contain any particular mathematical or statistical formula, but simply expresses the correlation between two variables, with the aim of making it visible in the form of an image. For this reason, it is impossible to minimize it.
Comment 7: same comments for tables 2 and a3 please ninimize the maths used-if possible spilt them or add some graphics
Response 7: Thank you for your comment. In table 2 we reported all the information that a linear regression require. These pieces of information are necessary for the readers to interpret results.
Regarding table A3, although we agree with You on its huge size, we do not see any excess of maths: it is a cross table for each variable in the row and the FN function in the immediate postoperative period and at 12 months. Here too, we were aware of the size of the table and we intentionally reported it in the appendix as supplementary material.
Comment 8: add more elements in discussion
Response 8: Thank you for your comment. Although quite general and vague as suggestion, we add more information regarding the role of biomechanical stiffness and its implication in the surgery.
Comment 9: add more and more recent references
Response 9: Since the aim of the study was to identify the role of consistency in vestibular schwannoma surgery, our discussion focused on everything that the English-language literature had to offer on this specific topic, without any time restrictions. We added few references on the role of Magnetic Resonance Elastography although its diffusion in brain imaging is still limited.
Comment 10: add more and more detailed conclusions
Response 10: see comment below
Comment 11: mention the innovantions of this study in the introduction
Response 11: By adding more context in the introduction, highlighting the contradictory results obtained by other studies that focused on radiological techniques such as MRE or radiological parameters such as ADC, it is easier to understand why we wanted to investigate the role of finding a radiological parameter that could correlate with the consistency of the lesion. This step is preliminary but essential in confirming an increase in intra-meatal pressure, the measurement of which will be the most objective factor in confirming this hypothesis and advancing a considered justification for early meatal opening.
Comment 12: add in the conclusion
Response 12: Due to the various limitations that have been expressly stated in the limitations paragraph, the conclusions remain largely hypothetical, with the aim in the immediate future being to improve certain methods (e.g., creating a three-dimensional region of interest) and expand the number of cases.
Reviewer 3 Report
Comments and Suggestions for Authors
This is a vualble work about the possibility of better defining a VS on preoperative brain MRI through an easily calculable parameter such as ADC thta could lead to an evolution of the classic dichotomy ‘solid vs cystic’. The association between increased consistency or lower N-ADCmin and worse immediate FN function ( based on the House-Brackmann score) may support the hypothesis of increased shock transmission to the facial nerve during dissection maneuvers. In addition to transmission or greater manipulation, one could also hypothesize a greater increase in intrameatal pressure. These data could have a relevant impact on the modulation of the surgical strategy.
The work is very interesting and well strucutred. nevertheless, no comment are presented by the authors concerning the role of the intra-operative monitoring of the facial nerve ( either with Cortico-bulbar potentials recording or with the direct stimulation of the facial nerve). This represents a relevant neurphysiological strategy to avoid FN impairment, where identification of the FN is as important as understanding the texture of the tumour... I would like to kindly ask the authors a comment about their strategy of intraoperative neurophysiological monitoring, that could aslo impact on the final clinical results. Thank you so much
Author Response
Comment 1: This is a vualble work about the possibility of better defining a VS on preoperative brain MRI through an easily calculable parameter such as ADC thta could lead to an evolution of the classic dichotomy ‘solid vs cystic’. The association between increased consistency or lower N-ADCmin and worse immediate FN function ( based on the House-Brackmann score) may support the hypothesis of increased shock transmission to the facial nerve during dissection maneuvers. In addition to transmission or greater manipulation, one could also hypothesize a greater increase in intrameatal pressure. These data could have a relevant impact on the modulation of the surgical strategy.
The work is very interesting and well strucutred. nevertheless, no comment are presented by the authors concerning the role of the intra-operative monitoring of the facial nerve ( either with Cortico-bulbar potentials recording or with the direct stimulation of the facial nerve). This represents a relevant neurphysiological strategy to avoid FN impairment, where identification of the FN is as important as understanding the texture of the tumour... I would like to kindly ask the authors a comment about their strategy of intraoperative neurophysiological monitoring, that could aslo impact on the final clinical results. Thank you so much
Response 1: Thank you for taking the time to review our manuscript and for your positive comments. We appreciate the opportunity to improve the quality of our work.
Considering your comment, We have moved back in the methods section to better highlight two references which described our surgical practice (one of these also includes a video).
We agree with you on the importance of IONM, so much so that in lines 171-174 we stated that often one of the main reasons that leads us to interrupt the removal of the tumor or to avoid insisting on dissecting the capsule from the nerve is precisely an excessive variation in neurophysiological parameters. The presence of continuous EMG discharges, a significant variation in corticobulbar potentials (which we try to repeat at regular intervals or at the end of each phase of tumor dissection), or a variation in direct stimulation parameters leads us to interrupt any maneuver, irrigate thoroughly with warm normal saline and/or use papaverine, either directly or with soaked absorbable hemostatic sponge, and wait 5-10 minutes. If we do not obtain an improvement in these parameters, we avoid continuing with the operation, convinced that the function of the facial nerve is more important than the extent of resection in preserving the patient's quality of life as much as possible. Our goal in the future will be to correlate this biomechanical stiffness with variations in neurophysiological parameters in a more consistent and systematic way.
We have better specified this aspect in the text (lines 174-178).
Round 2
Reviewer 2 Report
Comments and Suggestions for Authors
nice interesting but huge article; please minimize if possible
tables a1 and a3 are interesting but huge; please split them or add some graphics if possible
you need to add more and more recent references to support better the text
you need to add more radiological images
minimize or split table 2 as well
and explain better fig 2 and fig 4 -nice but huge work
may be better to split into 2 separate works
Comments on the Quality of English Languagethe english language can be improved
Author Response
Comment 1: nice interesting but huge article; please minimize if possible
Response 1: Thank you for taking the time to review our manuscript and for your positive comments. We add few improvements in the methods and results section (i.e., ROC analysis) as requested by reviewer #1 and a better specification of our use of intraoperative neurophysiological monitoring as requested by reviewer #3. The introduction has been expanded to provide a better context for the research, as you rightly suggested. Inevitably, this expansion has required a couple of additional comments in the discussion. We believe that reducing the information at this point would be a step backward compared to the first version of the manuscript or, in any case, would lead to a general reduction in information that would prevent the reader from understanding the aim and the mechanisms behind our study.
Comment 2: tables a1 and a3 are interesting but huge; please split them or add some graphics if possible
Response 2: We split tables a1 and a3 as suggested, dividing the information into radiological, intraoperative, and postoperative or clinical information. In the revised version you can find tables A1 and A2 for the distribution of radiological (table A1), intraoperative and clinical information (table A2) and the relationships of these radiological (table A4) and intraoperative (table A5) characteristics with FN outcome in postoperative period and at follow-up.
Comment 3: you need to add more and more recent references to support better the text
Response 3: we add more references especially on the comparison between cystic and solid vestibular schwanommas, considering especially those not included or published after the publication of the meta-analysis reported in the text (Wu, X.; Song, G.; Wang, X.; Li, M.; Chen, G.; Guo, H.; Bao, Y.; Liang, J. Comparison of Surgical Outcomes in Cystic and Solid Vestibular Schwannomas: A Systematic Review and Meta-Analysis. Neurosurg Rev 2021, 44, 1889–1902, doi:10.1007/S10143-020-01400-5).
The following references have been added: 3. Tang, I.P.; Freeman, S.R.; Rutherford, S.A.; King, A.T.; Ramsden, R.T.; Lloyd, S.K.W. Surgical Outcomes in Cystic Vestibular Schwannoma versus Solid Vestibular Schwannoma. Otol Neurotol 2014, 35, 1266–1270, doi:10.1097/MAO.0000000000000435; 4. Han, J.H.; Baek, K.H.; Lee, Y.W.; Hur, Y.K.; Kim, H.J.; Moon, I.S. Comparison of Clinical Characteristics and Surgical Out-comes of Cystic and Solid Vestibular Schwannomas. Otol Neurotol 2018, 39, e381–e386, doi:10.1097/MAO.0000000000001813; 5. Almefty, R.O.; Xu, D.S.; Mooney, M.A.; Montoure, A.; Naeem, K.; Coons, S.W.; Spetzler, R.F.; Porter, R.W. Comparison of Sur-gical Outcomes and Recurrence Rates of Cystic and Solid Vestibular Schwannomas. J Neurol Surg B Skull Base 2021, 82, 333–337, doi:10.1055/S-0039-1697039; 6. Zhang, L.; Ostrander, B.T.; Duhon, B.; Moshitaghi, O.; Lee, J.; Harris, M.; Hardesty, D.A.; Prevedello, D.M.; Schwartz, M.S.; Dodson, E.E.; et al. Comparison of Postoperative Outcomes in Cystic Versus Solid Vestibular Schwannoma in a Mul-ti-Institutional Cohort. Otology and Neurotology 2023, 45, 92–99, doi:10.1097/MAO.0000000000004062; 26. Zhou, L.; Wang, Z.; Hu, X.; Yang, D.; Zheng, Z. Comparison of Clinical Features and Surgical Outcomes of Cystic and Solid Vestibular Schwannoma. World Neurosurg 2025, 203, 124470, doi:10.1016/J.WNEU.2025.124470.
Comment 4: you need to add more radiological images
Response 4: Thank you for taking the time to review our manuscript. We add figure A1 in the Appendix A where 3 cases of VSs have been described.
Comment 5: minimize or split table 2 as well
Response 5: this table became table 3 in the revised version of the manuscript (see response 6). We have split this table as you suggested: table 3 for the multivariate linear regression for N-T2mean and table 4 for the multivariate linear regression of N-ADCmin
Comment 6: and explain better fig 2 and fig 4 -nice but huge work
Response 6: The captions of figure 2 and 4 have been completely revised in order to provide a better description of them. Furthermore, a table on the relationship between radiological and intraoperative characteristics with the consistency classes has been created (table 2 in the revised version of the manuscript).
Comment 7: may be better to split into 2 separate works
Response 7: thank you for pointing this out, we appreciate the possibility to express our point of view. We have increased the number of words by approximately 800 from the introduction to the conclusion compared to the first version of the manuscript, following the suggestions made by each reviewer while remaining within the 4,000 words required by the journal as a minimum requirement. We believe that the changes or the addition of other information have improved the content of the article. As for the work and its aim, with the current methods and results, we do not think it is worth dividing the work into two parts, leading to a loss of information necessary for a comprehensive understanding of the results. In any case, other lines of research on the same topic are currently underway, and we hope they will see the light of day in the near future.
Round 3
Reviewer 2 Report
Comments and Suggestions for Authors
this is a very nice and very
intretsing work-the authors made imporvements in all the fields-remainis a nice work with too many informations in the tables-please split the tables or some of them if possible by adding a graphic or some graphics may be- the tables are intresting but they are huge in order to underteand the meanings and the messages of this work
Comments on the Quality of English Language
the english language can be improved
Author Response
Comment 1: this is a very nice and very
intretsing work-the authors made imporvements in all the fields-remainis a nice work with too many informations in the tables-please split the tables or some of them if possible by adding a graphic or some graphics may be- the tables are intresting but they are huge in order to underteand the meanings and the messages of this work
Response 1: Thank you for taking the time to review our manuscript, for your positive comments, and for your suggestions. We decided to split further Tables A4 and A5, considering the relationships of radiological and intraoperative characteristics and the FN function in the immediate postoperative period (tables A4 and A5) and at 12-month follow-up (tables A6 and A7). We hope we have made the necessary changes, but we are happy to make further clarifications if felt necessary.